

# A retrospective study of surgical treatment and outcome among women with adnexal torsion in eastern Taiwan from 2010 to 2015

Ci Huang[1], Mun-Kun Hong[1], Tang-Yuan Chu[1] and Dah-Ching Ding[1,2]

[1] Department of Obstetrics and Gynecology, Hualien Tzu Chi Hospital, Buddhist Tzu Chi Medical Foundation, Tzu Chi University, Hualien, Taiwan, Taiwan
[2] Institute of Medical Sciences, Tzu Chi University, Hualien, Tawian, Taiwan

## ABSTRACT

**Background**. Adnexal torsion is a gynecologic emergency that requires surgical treatment. In this study, we reviewed the surgical outcomes of women with adnexal torsion in eastern Taiwan (Hualien county, area 4,629 $km^2$, 330,000 residents).

**Methods**. This retrospective study included 42 women diagnosed with surgically-proven adnexal torsion from January 1, 2010, to September 31, 2015. We compared the symptoms, objective findings, and surgical outcomes of patients who underwent laparotomy or laparoscopy.

**Results**. The laparoscopy and laparotomy groups included 27 and 15 patients, respectively. The most common symptom and sign was abdominal pain, followed by nausea and vomiting. In all patients, an adnexal tumor was detected through ultrasound. The median and range of time from admission to surgery was 1.5 (1–11.5) and 1.0 (1–11) hours in the laparotomy and laparoscopy groups, respectively. Compared with those undergoing laparotomy, the smaller tumor size [7 (4.2–10) vs. 10 (7–17) cm] and shorter hospital stay [4 (2–8) vs. 6 (3–9) days] in patients undergoing laparoscopy were significantly noted, respectively ($P < 0.01$). No differences were observed in age, operative time, and blood loss between both groups. The surgeries performed were mostly detorsion with cystectomy and adnexectomy. The most common pathology was a simple ovarian cyst, followed by teratoma. Regarding the surgical types, older age is the only risk factor for radical surgery.

**Discussion**. Acute onset of abdominal pain with a presenting ovarian tumor is the most common feature of adnexal torsion. Laparoscopic surgical group showed a small tumor size and a short ER hospital stay than laparotomy. Older age is the risk factor for radical surgery.

Corresponding author
Dah-Ching Ding,
dah1003@yahoo.com.tw

## INTRODUCTION

Adnexal torsion is a gynecologic emergency that requires surgical treatment (*Huang & Wang, 2011*; *Huang, Hong & Ding, 2017*). It is defined as twisting of the ovary, fallopian tube, or adnexal mass, inducing adnexal torsion. Partial or complete rotation of the

ovarian vascular pedicle obstructs venous outflow and arterial inflow (*Chang, Bhatt & Dogra, 2008*). Both benign and malignant lesions of the ovary may be the leading causes of adnexal torsion.

Surgical intervention is the gold standard for diagnosis and treatment of adnexal torsion. Conventionally, a twisted ovary or adnexa is excised completely (*Houry & Abbott, 2001*). However, adnexa-sparing surgery has emerged as an alternative (*Ding & Chen, 2005*; *Spinelli et al., 2013*; *Nair, Joy & Nayar, 2014*; *Ding & Chang, 2016*). Conservative surgery such as detorsion with cystectomy or cyst aspiration is preferred to preserve adnexal function. A previous study revealed that 51.4% of patients presenting to the emergency room were diagnosed with adnexal torsion (*Lo et al., 2008*).

Therefore, this study investigated the clinical characteristics of women with adnexal torsion in eastern Taiwan and compared the surgical outcomes of laparotomy and laparoscopy. We also calculate the risk factor for radical surgery.

## METHODS

This retrospective study analyzed the discharge data of women diagnosed with surgically proven adnexal torsion at Hualien Tzu Chi Hospital from January 1, 2010, to September 31, 2015. This study was approved by the Research Ethics Committee of Hualien Tzu Chi Hospital (IRB107-20-B). The Ethics Committee waived the need for informed consent from participants of this study.

All patients diagnosed with adnexal torsion had records of the International Classification of Diseases, Ninth Revision, Clinical Modification (ICD-9-CM) code 620.5 in their discharge notes. A total of 45 women were enrolled into our study, and 42 were diagnosed with adnexal torsion after surgery. Three patients did not receive surgical intervention because they refused surgical intervention, were lost to follow-up, or opted to receive medical treatment. Information on clinical characteristics, including age, medical history, operative time, and surgical methods, was obtained from patients' electronic medical records. Abdominal pain was defined as diffuse pain in the lower abdomen. In addition, surgical findings and pathological reports were obtained.

The surgical routes were laparotomy and laparoscopy. The surgical methods for the management of adnexal torsion were cystectomy, adnexectomy, and detorsion.

Statistical analyses were performed using SPSS version 25 (IBM, New York, NY, USA). All continuous variables are presented as median (range). All categorical variables are presented as numbers (percentage). Mann–Whitney $U$ test was used to compare the average of variables between two groups to determine the association between two continuous variables. Fisher's exact test was used to determine the difference between two categorical variables. Logistic regression was used to determine the odds ratio in the radical surgery group compared to the conservative surgery group. A $P$ value of $<0.05$ was considered statistically significant.

**Table 1 Preoperative symptoms and signs associated with adnexal torsion ($n = 42$).**

|  | Laparoscopy ($n = 27$) | Laparotomy ($n = 15$) | P-value |
|---|---|---|---|
| Ovarian or pelvic mass | 27 | 15 |  |
| Pelvic pain | 26 | 14 | 0.59 |
| Nausea and vomiting | 5 | 3 | 0.60 |
| Peritoneal sign | 1 | 1 | 059 |
| WBC > 12,000 | 4 | 2 | 0.63 |
| Fever | 0 | 1 | 0.35 |
| Urinary symptoms | 1 | 0 | 0.64 |
| Diarrhea | 0 | 2 | 0.15 |

Notes.
 P-value: Fisher's exact test.

**Table 2 Time interval between variables.**

|  | Laparoscopy ($n = 27$) Median (range) | Laparotomy ($n = 15$) Median (range) | P-value |
|---|---|---|---|
| From symptom onset to ED admission (hr) | 12 (0–36) | 24 (3–96) | 0.03[*] |
| From admission to surgery (hr) | 1.5 (1–11.5) | 1.0 (1–11) | 0.81 |
| From gynecologic evaluation to surgery (hr) | 2.0 (1–11) | 2.0 (1–11) | 0.85 |

Notes.
 P-value: Mann–Whitney $U$ test.
 ED, emergency department.
 *P-value < 0.05 was considered statistically significant after test.

# RESULTS

A total of 42 patients were surgically diagnosed with adnexal torsion during the study period. Table 1 illustrates the symptoms and signs of patients with adnexal torsion. The laparoscopy and laparotomy groups included 27 and 15 patients, respectively. In both groups, the most common symptom was lower abdominal pain (95.2%), followed by nausea and vomiting experienced by three and five patients in the laparoscopy and laparotomy groups, respectively. Regarding other symptoms, one patients in the laparoscopy group developed urinary symptoms, and two in the laparotomy group developed diarrhea. On examination, adnexal masses (100%) were noted in all patients. Four patients in the laparoscopy group and two in the laparotomy group had leukocytosis. One patient in the laparotomy group had fever, and one in each group had peritoneal signs. In our series, adnexal torsion was suspected in 64.4% of patients.

The median time (range) from symptom onset to seeking medical help was 12 (0–36) and 24 (3–96) hours in the laparoscopy and laparotomy groups, respectively ($p = 0.03$, Table 2). The median time (range) from admission to surgery was 1.5 (1-11.5) and 1.0 (1–11) hours in the laparoscopy and laparotomy groups, respectively. The median time (range) from gynecologic evaluation to surgery was 2.0 (1–11) and 2.0 (1–11) hours in the laparoscopy and laparotomy groups, respectively.

**Table 3  Pathological diagnosis of the adnexal tumor.**

|  | Laparoscopy ($n = 27$) | Laparotomy ($n = 15$) | P-value |
|---|---|---|---|
| Simple cyst | 13 | 4 | 0.15 |
| Mature cystic teratoma | 6 | 4 | 0.51 |
| Endometrioma | 4 | 1 | 0.40 |
| Fibroma or fibrothecoma | 0 | 4 | 0.01[*] |
| Necrosis of ovary | 1 | 0 | 0.64 |
| Complex cyst | 3 | 2 | 0.59 |

**Notes.**
P-value: Fisher's exact test.
*P-value $< 0.05$ was considered statistically significant after test.

Table 3 presents the pathological findings of adnexal torsion. The most frequent pathology was a simple cyst ($n = 17$, 13 and 4 in the laparoscopy and laparotomy groups, respectively), followed by mature cystic teratoma ($n = 10$, 6 and 4 in the laparoscopy and laparotomy groups, respectively). The remaining pathologies were endometrioma, fibroma, complex ovarian cyst and ovarian necrosis.

Table 4 provides a comparison of the surgical characteristics of patients who underwent laparotomy or laparoscopy. The median (range) age of patients was 31 (13–76) and 41 (12–82) years in the laparoscopy and laparotomy groups, respectively. The median (range) operative time was 64 (20–200) and 70 (45–168) minutes in the laparoscopic and laparotomy groups, respectively ($P = 0.50$). The median (range) blood loss was 50 (50–500) ml and 50 (50–400) ml in the laparoscopy and laparotomy groups, respectively ($P = 0.30$). The median (range) tumor size was 7 (4.2–10) and 10 (7–17) cm in the laparoscopy and laparotomy groups, respectively ($P < 0.01$). The median hospital stay was significantly shorter in the laparoscopy group than in the laparotomy group (4.0 vs. 6.0 days, $P < 0.01$). The total number of cases managed by cystectomy was 12, in which 10 and 2 cases were managed by laparoscopy and laparotomy, respectively ($P = 0.09$). The total number of cases managed by adnexectomy was 27, in which 15 and 12 cases were managed by laparoscopy and laparotomy, respectively ($P = 0.10$). Other two surgeries were drainage and fixation by laparotomy.

Table 5 compared the basic characteristics, surgical, and pathological parameters between conservative and radical surgery groups. We found the age was significant different between the both groups ($P = 0.001$). Older age was noted in radical surgery group. The other parameters was no difference between both groups.

Table 6 calculated the odds ratio between between conservative and radical surgery groups. We found older age is the risk factor for radical surgery [adjusted odds ratio (95% CI) $= 1.14$ (1.04–1.24), $p = 0.004$]. Patients with nausea or vomiting revealed a low risk for radical surgery (adjusted odds ratio (95% CI) $= 0.02$ (0.00–0.97), $p = 0.048$).

## DISCUSSION

In our series, 64.4% of patients were suspected to have adnexal torsion. In other reports, the incidence of suspected cases ranges from 18% to 62% before surgery (*Cohen et al.,*

**Table 4** Surgical characteristics in patients with adnexal torsion underwent laparoscopy and laparotomy.

| | Laparoscopy ($n = 27$) Median (range) | Laparotomy ($n = 15$) Median (range) | P-value |
|---|---|---|---|
| Age (year) | 31 (13–76) | 41 (12–82) | 0.26 |
| Surgical time (min) | 64 (20–200) | 70 (45–168) | 0.50 |
| Blood loss (mL) | 50 (50–500) | 50 (50–400) | 0.30 |
| Tumor size (cm) | 7 (4.2–10) | 10 (7–17) | <0.01[*] |
| Hospital stay (day) | 4 (2–8) | 6 (3–9) | <0.01[*] |
| Surgery[a] $n$ (%) | | | |
| Detorsion and drainage | 1 (3.7%) | 0 (0%) | 0.64 |
| Detorsion and cystectomy | 10 (37.0%) | 2 (13.3%) | 0.09 |
| Detorsion and fixation | 0 (0%) | 1 (6.6%) | 0.35 |
| Ovarian or adnexal resection | 15 (55.6%) | 12 (80.0%) | 0.10 |

**Notes.**

P-value: Mann–Whitney $U$ test.

[a] P-value: Fisher's exact test.

[*] P-value < 0.05 was considered statistically significant after test.

*2001*; *Houry & Abbott, 2001*; *Oelsner et al., 2003*). Suspicion of adnexal torsion is the key to early diagnosis (*Cohen et al., 2001*). Adnexal torsion should be suspected in patients with a history of an enlarged ovary or pelvic surgery complicated with abdominal pain (*Houry & Abbott, 2001*). Tubal ligation is the most frequent surgical history in 40% of cases with adnexal torsion (*Houry & Abbott, 2001*).

Table 7 compared the current study to the previous literatures. Abdominal pain accounted for the most presenting symptoms. Adnexal torsion was mostly happened at the Rt side. The tumor size was mostly between 5–10 cm. Laparoscopic surgery was the most surgical type. The proportion of conservative or radical surgery was variant in different studies. In our study, we found the older age is the risk factor for radical surgery. We speculated, for young patients, the conservation of ovary is important for their future fertility. Thus, conservative surgery is largely performed for young patients with ovarian torsion.

The study regarding fertility after ovarian torsion surgery is lacking. One report showed patients with one ovary (mostly by oophorectomy due to ovarian torsion or other pathology) did not affect the reproductive capability (*Lass, 1999*). But they have a shorter reproductive life span due to less primordial follicles (*Lass, 1999*).

Objective findings are variable and are rarely significant in patients with adnexal torsion. A study of 179 patients with adnexal torsion showed leukocytosis and fever in 20.1% and 7.8% of the patients (*Nair, Joy & Nayar, 2014*). In our study, one patient had a fever (2.3%) and six patients had leukocytosis (14.2%).

The sensitivity of pelvic ultrasound for diagnosing adnexal torsion ranges from 40% to 75% (*Mashiach et al., 2011*). Twisted adnexal masses are often midline and located anterior to the uterus; other findings include a cystic, solid, or complex mass with or without pelvic free fluid and thickening of the wall with cystic hemorrhage (*Mashiach et al., 2011*). A previous study found adnexal tumors larger than five cm in 88.4% of patients with adnexal

**Table 5 Surgical characteristics in patients with ovarian torsion underwent radical or conservative surgery.**

|  | Radical surgery (*n* = 27) Median (range) | Conservative surgery (*n* = 15) Median (range) | *P*-value |
|---|---|---|---|
| Age (year) | 42 (12–82) | 24 (13–45) | <0.01* |
| Surgical time (min) | 70 (29–190) | 60 (30–200) | 0.66 |
| Blood loss (mL) | 50 (50–500) | 50 (50–500) | 0.66 |
| Tumor size (cm) | 8.5 (4.2–17) | 7.0 (4.6–12) | 0.11 |
| Hospital stay (day) | 5 (2–9) | 4 (2–7) | 0.11 |
| Time interval |  |  |  |
| From symptom onset to ED admission (hr) | 12 (0–48) | 12 (1–96) | 0.76 |
| From admission to surgery (hr) | 1.5 (1–11) | 1 (1–11.5) | 0.84 |
| From gynecologic evaluation to surgery (hr) | 2 (1–11) | 2 (1–11) | 0.97 |
| Op type [*n* (%)] |  |  | 0.1 |
| Laparoscopy | 27 (55.6%) | 12 (80%) |  |
| Laparotomy | 12 (44.4%) | 3 (20%) |  |
| Tumor type (*n*) |  |  |  |
| Simple cyst | 9 (33.3%) | 8 (53.3%) | 0.17 |
| Mature cystic teratoma | 6 (22.2%) | 4 (26.7%) | 0.51 |
| Endometrioma | 4 (14.8%) | 1 (6.7%) | 0.4 |
| Fibroma or fibrothecoma | 4 (14.8%) | 0 (0%) | 0.15 |
| Necrosis of ovary | 1 (3.7%) | 0 (0%) | 0.64 |
| Symptoms and signs (*n*) |  |  |  |
| Ovarian or pelvic mass | 27 (100%) | 15 (100%) | 1 |
| Pelvic pain | 25 (92.6%) | 15 (100%) | 0.5 |
| Nausea and vomiting | 3 (11.1%) | 5 (33.3%) | 0.09 |
| Peritoneal sign | 2 (7.4%) | 0 (0%) | 0.4 |
| WBC > 12,000 | 5 (18.5%) | 1 (6.7%) | 0.53 |
| Fever | 1 (3.7%) | 0 (0%) | 0.39 |
| Urinary symptoms | 0 (0%) | 1 (6.7%) | 0.35 |
| Diarrhea | 1 (3.7%) | 1 (6.7%) | 1 |

torsion (*Nair, Joy & Nayar, 2014*). The whirlpool sign over the vessel pedicle has been reported to be a sign of adnexal torsion (*Vijayaraghavan, 2004*). In our study, the absence of blood flow showed low sensitivity for diagnosing adnexal torsion.

In the present study, the most common pathology was a simple ovarian cyst, followed by teratoma. Ashwal et al. also showed that the most common pathology was a simple cyst (*Ashwal et al., 2015*). However, *Lo et al. (2008)* found that teratoma was the most common pathology.

Multiple tumor markers, such as AFP (alpha-fetoprotein), CEA (carcinoembryonic antigen), HCG (human chorionic gonadotropin), and CA125, have been used for diagnosing ovarian neoplasms. CA125 is used for diagnosing epithelial ovarian tumors; however, it shows poor sensitivity and specificity (*Moss, Hollingworth & Reynolds, 2005*). In addition, CA125 levels are elevated in other benign conditions such as adenomyosis and

**Table 6  Factors associated with receiving oophorectomy ($n = 42$).**

| | Crude Odds Ratio (95% CI) | p value | Adjusted Odds Ratio (95% CI) | p value |
|---|---|---|---|---|
| Age | 1.10 (1.03 to 1.17) | 0.004[*] | 1.14 (1.04 to 1.24) | 0.004[*] |
| Simple cyst (Y vs. N) | 0.44 (0.12 to 1.59) | 0.210 | 0.34 (0.05 to 2.44) | 0.284 |
| Mature cystic teratoma (Y vs. N) | 0.79 (0.18 to 3.39) | 0.746 | | |
| Endometrioma (Y vs. N) | 2.44 (0.25 to 24.04) | 0.446 | 0.37 (0.02 to 8.23) | 0.532 |
| Fibroma or fibrothecoma (Y vs. N) | 1.05E9 (NA) | 0.999 | | |
| Necrosis of ovary (Y vs. N) | 9.32E8 (NA) | 1.000 | | |
| Symptoms and signs | | | | |
| Ovarian or pelvic mass (Y vs. N) | NA | NA | | |
| Pelvic pain (Y vs. N) | 0.00 (NA) | 0.999 | | |
| Nausea and vomiting (Y vs. N) | 0.25 (0.05 to 1.25) | 0.092 | 0.02 (0.00 to 0.97) | 0.048[*] |
| Peritoneal sign (Y vs. N) | 9.69E8 (NA) | 0.999 | | |
| WBC > 12,000 (Y vs. N) | 3.18 (0.34 to 30.16) | 0.313 | 42.03 (0.87 to 2,020.68) | 0.059 |
| Fever (Y vs. N) | 9.32E8 (NA) | 1.000 | | |
| Urinary symptoms (Y vs. N) | 0.00 (NA) | 0.999 | | |
| Diarrhea (Y vs. N) | 0.54 (0.03 to 9.28) | 0.670 | | |

**Notes.**
Data are presented as Odds ratio (95% CI).
*$P$-value < 0.05 was considered statistically significant after test.
Y, oophorectomy; N, no oophorectomy.

endometriosis (*Ghaemmaghami, Karimi Zarchi & Hamedi, 2007*). One case study reported the association between elevated CA125 levels and adnexal torsion (*McCarthy et al., 2010*). However, in an emergency setting, CA125 results cannot be rapidly obtained. Therefore, we did not routinely determine CA125 levels in our adnexal torsion cases.

C reactive protein (CRP) has been reported as a novel marker of adnexal torsion (*Tobiume et al., 2011*; *Damigos, Johns & Ross, 2012*; *Bakacak et al., 2015*; *Bolli et al., 2017*). In an animal experiment, after adnexal ischemia, the plasma CRP level was found to be elevated (*Bakacak et al., 2015*). In a previous study, a CRP level of <0.3 mg/dL was correlated with a favorable prognosis of ovarian conservation in adnexal torsion (*Tobiume et al., 2011*). However, in our study, we did not routinely evaluate CRP levels.

The main reason for surgical intervention in cases of adnexal torsion is lower abdominal pain and enlarged adnexal masses (*Cohen et al., 2001*). In our patients, the time from admission to surgery was 3 h, which was shorter than the time reported by Ashwal of 4.6 h (*Ashwal et al., 2015*).

Huang et al. (2018), *PeerJ*, DOI 10.7717/peerj.5995

**Table 7  Comparison of present study to the previous studies.**

| | Our study | Nair, Joy & Nayar (2014) | Vijayalakshmi et al. (2014) | Spinelli et al. (2013) | Vijayaraghavan (2004) |
|---|---|---|---|---|---|
| No. of patients | 42 | 70 | 18 | 30 | 21 |
| Age | 12–82 yrs | 11–91 yrs | 25–72 yrs | 2 months–18 yrs | 7–69 yrs |
| Pregnancy | 6.6% | 2.9% | | | 4.7% |
| Abdominal pain | 95.2% | 95.7% | 77.8% | 100% | 100% |
| Nausea/vomiting | 19% | 65.7% | 27.8% | 56.7% | |
| Fever | 2.3% | 12.9% | 5.6% | 20% | |
| Leukocytosis | 14.2% | 44% | | 63.3% | |
| Doppler diagnosis of torsion | | 25.7% | | 63% | 95.2% |
| Free fluids in pelvis | | 23.8% | | 26.7% | |
| Right side | 51.1% | 55.7% | 50% | 70% | |
| Left side | 40.0% | 42.9% | 38.9% | 30% | |
| Bilateral | 2.2% | 1.4% | 11.1% | | |
| Size <5 cm | 11.1% | | | | |
| Size 5~10 cm | 71.1% | 71.4% | 33.3% | | |
| Size >10 cm | 17.7% | 11.4% | 44.4% | | |
| Mature cystic teratoma | 23.8% | 22.8% | 16.7% | 16.7% | 28.5% |
| Mucinous cystadenoma | | 7.1% | 33.3% | 3.3% | 9.5% |
| Serous cystadenoma | | 15.7% | 16.7% | | 47.6% |
| Hemorrhagic necrosis | 2.3% | 30.4% | | | 9.5% |
| Laparoscopy | 64.2% | 81.4% | | 40% | |
| Laparotomy | 35.8% | 18.6% | | 60% | |
| Conservative surgery | 35.8% | 54.3% | | 46.7% | |
| Radical surgery | 64.2% | 45.7% | | 53.3% | |

In our study, no significant difference was observed in operative time between the laparoscopy and laparotomy groups. *Lo et al. (2008)* also found no significant difference in operative time between laparoscopy and laparotomy groups.

The only significant factor for choosing surgical route is tumor size (*Lo et al., 2008*). In our series, the mean diameter of adnexal tumors was 6.9 and 10.4 cm in the laparoscopy and laparotomy groups, respectively ($P < 0.01$). Significantly smaller ovarian cysts were noted in the laparoscopy group than in the laparotomy group.

Although adnexal torsion occurs in all age groups, it is most commonly observed in reproductive age groups (*Tsafrir et al., 2012*; *Vijayalakshmi et al., 2014*). In this study, most patients were postmenarchal, and eight women were postmenopausal.

In our study, younger patients received laparoscopy (33.6 vs. 41.0 years, $P = 0.18$). A previous study also showed no difference in age between the laparoscopy and laparotomy groups (30 vs. 32 years). Cosmetic reasons may affect the chosen surgical route such as minimal invasive laparoscopy (*Lee et al., 2010*). The other reason is that ovarian malignancy is more often suspected in older patients, who may choose laparotomy (*Moorman et al., 2008*).

Significantly shorter length of hospital stay was noted in the laparoscopy compared to the laparotomy group (4.0 vs 5.4 days, $p < 0.01$). This observation is consistent with the previous reports (*Lo et al., 2008*; *Grammatikakis et al., 2015*).

We have shown that laparoscopic detorsion and cystectomy are feasible in adults and pregnant women with adnexal torsion (*Ding & Chen, 2005*; *Ding & Chang, 2016*). Detorsion carries no risk of thrombosis (*Ashwal et al., 2015*; *Huang, Hong & Ding, 2017*). In recent years, detorsion without cystectomy or adnexectomy has become more prevalent (*Ashwal et al., 2015*).

In a previous study on a patient with a benign tumor, the potential for ovary conservation was found to be greater in adnexal torsion cases with a short period from the onset of abdominal pain to surgery (*Tobiume et al., 2011*). In our study, we performed detorsion with ovarian cystectomy irregardless of tumor size. Moreover, a benign ovarian pathology should be obtained during surgery (*Jeon et al., 2017*).

Previous studies have shown that ovarian morphology and function are restored to normal after detorsion (*Galinier et al., 2009*; *Geimanaite & Trainavicius, 2013*; *Parelkar et al., 2014*). Moreover, studies have suggested that conservative treatment of adnexal torsion is safe. After adnexal detorsion, menstruation and normal ovarian morphology were restored in all our premenopausal patients.

The recurrence rate after detorsion is remained unknown. One report revealed 63% and 8.7% recurrence rates in the twisted normal adnexa and twisted abnormal adnexa, respectively (*Pansky et al., 2007*). There are several methods could prevent recurrence including oophoropexy (*Pansky et al., 2007*) and oral contraceptives (*Fee, Kanj & Hoefgen, 2017*). However, both methods lack long term follow-up study.

## CONCLUSIONS

In conclusion, the early diagnosis and intervention of adnexal torsion are crucial. Acute onset of abdominal pain with a presenting adnexal tumor is the most common feature

of adnexal torsion. Laparoscopic surgical group showed a small tumor size and a shorter hospital stay than laparotomy. Older age is the risk factor for radical surgery.

## ACKNOWLEDGEMENTS

The authors thank Dr. Jon-Son Kuo for English editing. We thank Wallace Academic Editing group for polishing the English.

### Funding

The authors received no funding for this work.

### Competing Interests

The authors declare there are no competing interests.

### Author Contributions

- Ci Huang performed the experiments, analyzed the data, prepared figures and/or tables, authored or reviewed drafts of the paper, approved the final draft.
- Mun-Kun Hong and Tang-Yuan Chu contributed reagents/materials/analysis tools, approved the final draft.
- Dah-Ching Ding conceived and designed the experiments, performed the experiments, analyzed the data, prepared figures and/or tables, authored or reviewed drafts of the paper, approved the final draft.

### Human Ethics

The following information was supplied relating to ethical approvals (i.e., approving body and any reference numbers):

This study was approved by the Research Ethics Committee of Hualien Tzu Chi Hospital (IRB107-20-B).

### Data Availability

The data is included in the results section and Supplemental Files.

### Supplemental Information

Supplemental information for this article can be found online at http://dx.doi.org/10.7717/peerj.5995#supplemental-information.

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
