# Peer review of "A retrospective study of surgical treatment and outcome among women with adnexal torsion in eastern Taiwan from 2010 to 2015"

_PeerJ, doi:10.7717/peerj.5995_

## Round 0.1 · original submission · Major Revisions

· Academic Editor

Major Revisions

Dear Authors,

The Reviewers found your manuscript very interesting.

Reviewer #2 and Reviewer #3 raised serious concerns, while Reviewer #1 was more positive and generally in favor of publication after a minor revision.

In light of the Reviewer's comments, I personally evaluated the manuscript and consider that, after responding to the comments of Reviewer #1, I would suggest to consider the issues raised by Reviewer #2 and Reviewer #3, since they are critical and justified.

Please answer all comments in a formal rebuttal letter, discuss and incorporate them in the revised version of your manuscript in order to reach the standard requested for publication.

With personal regards

Salvatore Andrea Mastrolia, MD
PeerJ Academic Editor

·

Basic reporting

Dear Editor,
Thank you for the opportunity to review the interesting manuscript entitled ‘A retrospective study of surgical treatment and outcome among women with
adnexal torsion in eastern Taiwan from 2010 to 2015’.

The authors reviewed the outcomes of 42 female patients with adnexal torsion managed in eastern Taiwan between 2010 - 2015.
The paper is interesting and has merits to be accepted after minor revision.

Experimental design

Major concerns:
- Methods: Please use non-paratemtric tests or otherwise prove that data are normal distributed using a statistical test (Kolmogornov Smirnov for example). Please report the data as median median and range or IQR if they are not normal distributed. Chi-sure test should be Fischer s test in some cases - please detail.

Validity of the findings

Minor concerns:
In Abstract - please detail more clear what mean eastern Taiwan.
In Abstract - pleas present the data as median and range or IQR.
Table 5 should be moved to Discussions section, not Results.
Please discuss the recurrence of the torsion after surgical treatment in the literature and the fertility rate after these surgical procedures.

Reviewer 2 ·

Basic reporting

This study -A retrospective study of surgical treatment and outcome among women with adnexal torsion in eastern Taiwan from 2010 to 2015 is an importent issue among gynecology emergency.
Unfortunately this study cohort is small and cast represent truly the study subject.
The main idea is important and well disigned but again too small.

Experimental design

well desigh

Validity of the findings

There is no innovation in this study

Additional comments

When you have a small cohort -try to fined a special point to designate it from other big studeis

Reviewer 3 ·

Basic reporting

Comments to PEERJ 29482 entitled A retrospective study of surgical treatment and outcome among women with adnexal torsion in eastern Taiwan from 2010 to 2015

The authors retrospectively evaluated women who underwent surgical management for ovarian torsion and finally a total of 42 women were enrolled into the current study.
Among these, 27 women received laparoscopic approach and the remaining 15 underwent exploratory laparotomy. Conservative surgery was done in one-third of patients (35.8%), suggesting that the majority of patients with ovarian torsion were ended by oophorectomy. Although the authors did not comment this part, it is rational to be explained by only 64.4% of patients who had supposed ovarian torsion preoperatively. However, it is interesting to find that the time interval between the onset of symptom and the final operation is not long, suggesting that the ability of urgent operation is relatively good in the authors’ institute. Some comments are shown below.
1. it is relatively useful to read the difference between radical surgery and conservative surgery. If possible, the audience might be much more interested in this topic. I am wondering why the authors separate their patients into two groups based on the different operation approaches. Tumor size was significantly larger in the laparotomy group, and simple cyst was dominant in the laparoscopy group. Furthermore, fibroma and fibrothecoma were only found in the laparotomy group. All suggested that selection of operation might be biased. In addition, did it mean the surgical approach might be dependent on the “fear” of the malignancy? If the answer is yes, what was the value of the comparison between laparoscopy and laparotomy?
2. If four cases of fibroma and/or fibrothecoma are noted, did ascites exist?

Experimental design

Comparison between radical surgery and conservative surgery might be much more valuable.

Validity of the findings

If possible, please evaluate the relationship between parameters and radical surgery and provide the risk estimation for the radical surgery.

Annotated reviews are not available for download in order to protect the identity of reviewers who chose to remain anonymous.

---

## Round 0.2 · Minor Revisions

· Academic Editor

Minor Revisions

Dear Authors,

the Reviewers are favorable to the publication of your manuscript in PeerJ after a minor revision.

Please incorporate the suggested changes and submit a revised version of your manuscript in order to achieve publication.

Best regards

Salvatore Andrea Mastrolia
Peerj Academic Editor

·

Basic reporting

The authors made all the suggested modifications.
However, in Tables 2, 3,4 contrary to the results section, the authors did not report the data as median and range (still mean and std dev). Please change that.
I recommend the article to be published after this minor change.

Experimental design

No comment

Validity of the findings

No comment

Additional comments

In Tables 2, 3,4 contrary to the results section, the authors did not report the data as median and range (still mean and std dev). Please change that.

Reviewer 3 ·

Basic reporting

Clear and well written

Experimental design

acceptable

Validity of the findings

age is the final dependent factor for radical surgery, suggesting that complete family might result in the radical surgery by authors.

Additional comments

significant improvement and acceptable

---

## Round 0.3 · accepted · Accept

· Academic Editor

Accept

Dear Authors,

I would like to compliment with you for the efforts provided in addressing the Reviewers' comments.

All Reviewers felt that your manuscript has reached the level of publication and can be accepted in its current form.

Best regards

Salvatore Andrea Mastrolia
PeerJ Academic Editor

·

Basic reporting

The authors made all the suggested modifications. I recommend the paper to be published in the current form. Congratulations!

Experimental design

No comments.

Validity of the findings

No comments.

Additional comments

No comments.